# Impact Fracture Surfaces as the Indicators of Structural Steel Post-Fire Susceptibility to Brittle Cracking

**DOI:** 10.3390/ma16083281

**Published:** 2023-04-21

**Authors:** Mariusz Maslak, Michal Pazdanowski, Marek Stankiewicz, Anna Wassilkowska, Paulina Zajdel, Michal Zielina

**Affiliations:** 1Chair of Bridge, Metal and Timber Structures, Faculty of Civil Engineering, Cracow University of Technology, 31-155 Cracow, Poland; mstankiewicz@pk.edu.pl (M.S.); paulina.zajdel1@pk.edu.pl (P.Z.); 2Chair for Computational Engineering, Faculty of Civil Engineering, Cracow University of Technology, 31-155 Cracow, Poland; michal.pazdanowski@pk.edu.pl; 3Department of Water Supply, Sewerage and Environmental Monitoring, Faculty of Environmental Engineering and Energy, Cracow University of Technology, 31-155 Cracow, Poland; anna.wassilkowska@pk.edu.pl (A.W.); michal.zielina@pk.edu.pl (M.Z.)

**Keywords:** steel microstructure, post-fire properties, fracture surface, brittle cracking, impact test, shear fracture appearance, toughness

## Abstract

The results of experimental research on forecasting post-fire resistance to brittle failure of selected steel grades used in construction are presented and discussed in this paper. The conclusions are based on detailed analysis of fracture surfaces obtained in instrumented Charpy tests. It has been shown that the relationships formulated based on these tests agree well with conclusions drawn based on precise analysis of appropriate *F–s* curves. Furthermore, other relationships between lateral expansion *LE* and energy *W_t_* required to break the sample constitute an additional verification in both qualitative and quantitative terms. These relationships are accompanied here by values of the *SFA_(n)_* parameter, which are different, depending on the character of the fracture. Steel grades differing in microstructure have been selected for the detailed analysis, including: *S355J2+N*—representative for materials of ferritic-pearlitic structure, and also stainless steels such as *X20Cr13*—of martensitic structure, *X6CrNiTi18-10*—of austenitic structure and *X2CrNiMoN22-5-3* duplex steel—of austenitic-ferritic structure.

## 1. Introduction

In order to reliably evaluate the suitability of structural steel for potential service in the bearing structure after undergoing an episode of rapid heating followed by more or less prolonged holding it at raised temperature in fire conditions and, finally, effective cooling down, one has to verify whether, and if so to what extent, such steel has preserved its plastic properties, conditioned by sufficient ductility resulting in post-fire resistance to brittle cracking [1]. Subjecting any material to unintentional action of relatively high temperature of uncontrolled time profile always results in structural changes of weakening and sometimes even degrading character [2,3]. Changes of this kind usually are permanent, so they remain in the material after its complete cool down [4,5]. However, in many situations those changes may remain unnoticed, especially when post-fire technical condition evaluation of the building object is limited to only visual inspection of its bearing structure accompanied by often very cursory inventory of observed deformations. In the Authors’ opinion, the resistance of structural steel to brittle cracking is of key importance here. In particular, it is not so much about the a posteriori observation of the complete lack of susceptibility to initiate brittle cracks in the tested material, but about the observed post-fire loss of the capability to effectively arrest the unrestrained growth of such cracks.

This paper constitutes a continuation of previous works published earlier in [6,7,8]. In our opinion, it has been shown there that a typical set of post-fire tests pertaining to the determination of mechanical properties of structural steel after a fire incident should be mandatorily broadened by adding impact tests. If possible, these tests should be instrumented [9,10,11,12,13], with a hammer of sufficiently high potential energy, using appropriate data transmission and processing following the recommendations of the codes [14,15,16,17]. The applicability of these tests to forecasting structural steel resistance to brittle cracking has been proven many times (for example in [18]), though most of such tests did not refer to material cooled down after undergoing a fire incident. However, the results obtained allowed for reliable verification of the risk of initiating new cracks in the material, and then for assessing the degree of limiting their possible propagation [19,20,21,22]. Mechanisms of ductile failure under load have been recognized and identified as a result of such research. This seems to be crucial for the deliberations presented in this paper [23,24,25,26,27,28]. The quantitatively significant risk of brittle failure in prolonged use revealed during research in our opinion should constitute an important factor in the decision on disqualifying analyzed steel from the extended service, or at least on restricting the user-accepted scope and conditions of such service, as this brittle failure in structural steel components usually occurs abruptly, with no previous signs of progressive material degradation over time.

In the deliberations presented in this paper, we will focus on the description and interpretation of various impact fracture types obtained during our research on several steel grades. In our opinion, the morphology of such fractures, observed in practice, seems to clearly indicate the forecast post-fire susceptibility of a given material to brittle fracture under external load. The research on correlations of this type, related to structural steels effectively cooled down after an action of fire temperature, is, so far, relatively infrequent [29,30,31,32,33,34,35]. The interest of researchers seems to have been concentrated on registration of permanent changes in mechanical properties of tested steel grades, in particular their yield limits, ultimate strength, and modules of linear elasticity. Detailed insight into the mechanisms affecting the brittle or partially plastic form of initiating and propagating cracks observed after a fire incident, correlated with verification of the fracture surface obtained, has to be unequivocally referred to the conditions accompanying fracture development, as these mechanisms may be different when accompanying quasi static tensile test, and different in the case of the dynamic Charpy impact test.

Several grades representative of various types of structural steels have been selected for detailed analysis. These steel grades differ not only in scope of application but also in chemical composition, and thus in internal microstructure. It has been acknowledged that those characteristics should determine the expected mechanical properties identified on these steels after cooling down in fire episodes. A conventional low alloy *S355J2+N* steel [36], representative for a whole group of steels exhibiting ferritic-pearlitic structure (including among others the Chinese *X80* steel [37,38,39], and also many high strength steels [40]) was selected at the beginning. The results obtained for this steel were juxtaposed with the results characterizing the post-fire properties of selected stainless steels such as *X20Cr13* steel—of martensitic structure [41], *X6CrNiTi18-10* steel—of austenitic structure, and *X2CrNiMoN22-5-3* duplex steel—of austenitic-ferritic structure [42,43,44].

## 2. Preparation of Samples

Prior to the impact tests, the samples were subjected to thermal treatment of a formalized course, corresponding to the steady-state heating regime. This treatment simulated the action of fully developed fire on the sample. During the first phase, the samples were heated at a constant speed of 100 °C/min up to 600 °C (first series) or 800 °C (second series), then kept for 60 min at this temperature (Figure 1). After heating, the samples were cooled down to ambient temperature. For comparative purposes, two cooling scenarios were applied, namely slow cooling in the furnace to simulate the self-extinguishing of a fire and rapid cooling in water mist to simulate the fire extinguishing action of a fire brigade.

The heating temperature levels indicated above were selected intentionally, as it was preliminarily assumed that the first one during 60 min long exposure time would prove to be too low, while the second one would be sufficiently high, to initiate structural changes of permanent character in the tested steels.

The impact toughness tests were conducted at +20 °C to simulate summer conditions and at −20 °C to simulate winter conditions of the post-fire service of the tested steels.

Thus, the research encompassed 40 quantitatively different cases. For each of the considered steel grades and each of both toughness test temperature levels, four independent test cases related to the steel cooled down after a simulated fire episode (two heating temperature levels multiplied by two different cooling scenarios), and one so-called reference case for the steel in as-delivered condition (i.e., unaffected by the simulated fire action) was considered. Each test case was statistically analyzed on six independent steel samples to assure sufficient reliability of the obtained estimates. This means that 40 × 6 = 240 independent impact strength tests were conducted and interpreted in detail.

The final results were archived using a three-digit key to distinguish separate cases as shown in Table 1. The cases denoted by single digits, i.e., 1, 2, 3, and 4, respectively, refer in this key to the six element set of the so-called reference cases, obtained after testing the samples made of the material remaining in as-delivered condition (i.e., the material which was unaffected by the simulated fire action).

The juxtaposition of all the tested samples, with their descriptions, is depicted in Figure 2 (only the top layer of a three-tier set is visible there). The noteworthy difference in colors of the scale appearing on the surface of individual groups is induced by the differences in chemical composition of the corresponding steel grades. This scale observed on the surface of samples made of *S355J2+N* steel was only loosely bound to the substrate, while on all the stainless steel grades, it formed a tight surface layer.

## 3. Description of the Impact Strength Tests Conducted

The impact strength tests (Figure 3) of the samples were conducted following the recommendations of the codes [14,15,16,17] using instrumented Charpy hammer of the JB-W450E-L type (Figure 4a), of 450 J potential energy. The R8 beater (of American type) was selected, under the assumption that in construction, an impact by an object of more flattened surface is more probable (as compared to the classic R2 beater of European type) [45,46]. The hammer was equipped with a transducer to measure the force applied to the sample, and the accompanying force application point displacement was registered automatically with an encoder. Signals generated by both devices were gathered and processed by a data logger of high sampling frequency and subsequently analyzed by dedicated computer program. The results of each test were illustrated on automatically generated graphs, showing force, breaking energy, and displacement of force application point as functions of time, or alternatively force and breaking energy as functions of force application point displacement. A computer program automatically marked on these graphs the location of characteristic limit points as well.

For all test cases referring to the *S355J2+N* steel, as well as for those referring to stainless steel grades *X20Cr13* and *X6CrNiTi18-10*, full size impact test samples (ISO Charpy V-10) were used. However, in the cases pertaining to the *X2CrNiMoN22-5-3* stainless steel, the sample size had to be changed to ISO Charpy V-7.5 [47], as the energy of 450 J proved to be insufficient to break the full size sample in its as-delivered condition or a full size sample heated for one hour at 600 °C and subsequently cooled in water mist.

The notch profile testing of reference samples (i.e., of those made of steel in the as-delivered condition, unaffected by thermal treatment simulating the fire action) was conducted on the notch profile projector XT-50 (Figure 4b). It has been shown that these notches were located just below the upper tolerance limit specified in the code [14]. Analogous verification of samples cooled down after prior exposure to fire temperature showed an increase in the size of notches caused by the removal of scale developed as a result of high temperature oxidation resulting in several cases in notches exceeding the size tolerance limits prescribed by the codes. According to the provisions of the code [17], discrepancies of this type may occur in the impact strength tests, resulting in the brittle fracture of a sample overestimating the experimentally determined breaking energy by only 2 ÷ 3 J. However, should the ductile fracture be observed, this overestimation would be negligible. Therefore, in the following considerations, it was assumed that the measurement error induced by this factor would not be taken into account when dealing with post-fire susceptibility of considered steel grade to brittle fracture.

## 4. Parameters of the Observed Impact Fractures Used to Draw Conclusions

A *F–s* force–displacement graph obtained for a given fracture was used as a basis for the following considerations. In particular, these curves were qualified as category *E* (Figure 5a) or as category *F* (Figure 5b) according to the classification listed in the code [15]. This corresponds to the categories *B* and *C* listed in the code [16], respectively.

The general yield force *F_gv_* on these graphs corresponds to the initiation of yield at the developing fracture, while maximum force *F_m_* indicates the global maximum on the *F–s* curve. Unstable growth of the fracture is initiated at *F_in_* (force at the initiation of unstable crack propagation). This phase of the process is correlated with abrupt reduction of the applied force accompanied by minimal increase in the displacement of force application point. It is concluded when displacement corresponding to the force *F_a_* (force at the arrest of unstable crack propagation) is reached. At this moment, the effective fracture arrest begins and the sample undergoes plastic fracture. Let us note that the shape of the *F–s* graph in Figure 4b does not reveal the force *F_in_*. This means that the graphs of this type, qualified as category *F* (according to [15]), are to be correlated with a fracture exhibiting only completely stable phases. This means, in turn, that self-arrest of the fracture is fully effective. In the case of a *F–s* relationship belonging to the type *E* (according to [15]), as shown in Figure 5a, the capability of the sample to self-arrest an unstable growth of a fracture initiated during the preceding stage is higher the bigger the area under the experimentally obtained curve, provided that such an area is determined only for the displacement larger than the one associated with the limit force *F_a_*. This results in a relatively bigger area of the plastic breakthrough observed on the surface of the fracture. Let us note, also, that when *F_m_ = F_in_* holds, then the given *F–s* relationship is qualified as belonging to the category *C* or *D* in the sense of recommendations contained in [15], according to the character of the final phase of the fracturing process, determined by the way the force *F_a_* is revealed.

In the previous paper [7], we showed how the post-fire brittleness of the material is assessed based on the shape of the *F–s* curve obtained during the impact strength test under the assumption of specific (summer or winter) weather conditions. The evaluated pattern has to be related to the analogous pattern associated with the limit value of the fracturing energy *W_t,min_* = 27 J, for which the ductile-to-brittle transition temperature (DBTT) is defined. The steels, for which at a given temperature the experimentally obtained fracturing energy *W_t_ < W_t,min_*, would exhibit a tendency to dominantly brittle fractures. Thus, these steels would be incapable of effectively arresting micro-cracks initiated in their microstructure and growing in time. Therefore, such steels may not be recommended for further application in the construction industry. Only the steels for which the relation *W_t_* > *W_t,min_* is proven exhibit the capability to self-arrest the micro-crack growth [48]. In each of the tested cases, the surface limited from above by the experimentally obtained *F-s* curve and from below by the horizontal axis of the graph represents the energy *W_t_* required to fracture the sample.

When the experimentally obtained *F–s* curve does not show the steeply sloped part, as for instance happens in the case of the sample classified in the category *F* (Figure 5b), it means in general that the share of plastic fracture area is equal to 100% of the total fracture area. However, should such a segment be identified for a given fracture, then the approximate value of the ratio of ductile fracture surface may be estimated by applying one of the following formulae [15,49,50]:(1)$$SFA_{\left(1\right)}=\left[1-\frac{F_{in}-F_{a}}{F_{m}}\right]\cdot 100\%$$
(2)$$SFA_{\left(2\right)}=\left[1-\frac{F_{in}-F_{a}}{F_{m}+\left(F_{m}-F_{gy}\right)}\right]\cdot 100\%$$
(3)$$SFA_{\left(3\right)}=\left[1-\frac{F_{in}-F_{a}}{F_{m}+0.5\cdot \left(F_{m}-F_{gy}\right)}\right]\cdot 100\%$$
(4)$$SFA_{\left(4\right)}=\left[1-\sqrt{\frac{\frac{F_{gy}}{F_{m}}+2}{3}\left(\frac{\sqrt{F_{in}}}{\sqrt{F_{m}}}-\frac{\sqrt{F_{a}}}{\sqrt{F_{m}}}\right)}\right]\cdot 100\%$$

The abbreviation *SFA*_(*n*)_ (%) denotes here the shear fracture appearance. The higher its value, the more resistant the considered material is to brittle failure.

The lateral expansion *LE* (mm) was measured on each sample using the gauge depicted in Figure 4c. The idea of this measurement is depicted in Figure 6. As shown there, this expansion occurs only on at least partially ductile fractures. Polynomial regression formulae relating this parameter to the corresponding sample fracturing energy are commonly known [51,52]. These formulae indicate that in the case of ferritic steels, the capability of the material to absorb dynamic energy, here expressed as the fracturing energy *W_t_*, may not be extended indefinitely. At high impact energy, a significant reduction of ductility measured via the *LE* parameter occurs for such steels. However, in general, a higher *LE* value, at the fracture energy values typical for the structural steels, should be associated with higher resistance to brittle failure exhibited by the test samples. Thus, the *LE* parameter in this sense represents the highest relative increase in sample width after brittle failure, determined in the cross section directly hit by the hammer (annex B to the code [14]). In this cross section, the freedom to realize plastic deformations resulting in permanent deformation of its initial rectangular shape at coincident propagation of the fracture is the highest. Detailed analysis of these deformations exhibited by samples cooled down after exposure to fire temperature and later subjected to impact fracture tests constitutes one of basic premises leading to conclusions regarding post-fire impact strength of a given steel grade. In order to do so, one has to identify particular zones associated with subsequent phases of the fracture on the typical fracture surface of mixed ductile (meaning partly plastic and partly brittle) character. These zones are depicted in Figure 7. As may be seen, the fracture is initiated directly under the notch, in spite of the fact that the hammer hits the other side of the sample, since the local stress concentration zone develops directly adjacent to the notch. On the side surfaces of the sample, the number of the degrees of freedom in strains is higher than in the center, and thus in those areas, plastic fracture may occur. The central part of the sample is dominated by an unstable brittle fracture. This manifests itself by an area of laminated fracture in the center of the sample.

Visual observation of the fracture of mixed type allows for simplified determination of the *SFA*_(*n*)_ parameter (annex C to the code [14]), yielding the value of *SFA*_(5)_ (%) as a result. One has to determine the percentile ratio of the laminated fracture area of brittle character (the so-called flat fracture region), indicated by the dimensions A and B in Figure 8, with respect to the full cross section area. The shear area of the fracture surface will therefore complement this value to 100%.

More accurate methods used to determine this parameter, based on visual observation of the fracture area, apply advanced image processing approaches [53,54].

## 5. Microstructural Studies of Impact Fractures

The microstructure of each impact fracture surface obtained in our experiment was analyzed using a scanning electron microscope (SEM) Hitachi S-3400N VP-SEM (Figure 9). This device may be set up to work under both high vacuum conditions and variable pressure. It typically yields magnifications of 10 to 20,000 times, with a very good surface contrast and depth of field for resolutions of 4 to 10 nm. To perform the chemical analysis, the microscope described above was equipped with an energy dispersive spectroscopy (EDS) *ThermoScientific* Si(Li) detector with a resolution of 135 eV and with the Noran System 7 (NSS) analytical software.

The SEM works on the principle of generating an electron beam in the thermo-emission gun. This beam, while passing through the electromagnetic lens system, hits the tested sample placed in a vacuum chamber. Depending on the accelerating voltage of the primary beam (from 0.3 kV to maximum 30 kV), the electrons bombarding the sample surface penetrate the tested material to a depth of several micrometers. Secondary (SE) and backscattered (BSE) electrons, as well as characteristic X-rays, are then emitted from the excited volume of a sample. Those signals are used to image the sample surface topography (SE detector), to reveal the contrast of the atomic number of the tested sample (BSE detector), and to identify the micro-sample elemental composition (EDS detector) [55,56].

Detailed results pertaining to the identification and qualitative interpretation of permanent changes observed by the authors of this paper in the microstructure of considered steel grades after action of simulated fire episodes following various development scenarios have been published and discussed in paper [8]. Therefore, these results will be called upon in the following deliberations only in the scope required for proper commenting of the results referring to the same steels, but obtained during the research of their post-fire impact strength, reported below.

In energy dispersive spectroscopy (EDS), determination of the qualitative and then quantitative composition is performed using X-ray excitation parameters adequate to the elements expected in the composition of the sample [57]. In the semi-quantitative EDS, analysis element concentrations are normalized to 100%, seemingly yielding an impression of error-free analysis. Correction of quantitative ratios is executed automatically by application of specialized Noran System 7 software. In particular, the correction by the *Z* factor results from the influence of the atomic number on the X-ray excitation efficiency, the correction by the *A* factor takes into account the probability of X-ray absorption by the sample, the correction by the *F* factor is associated with the possibility of additional emission of energy quantums induced by characteristic and continuous X-rays. It is difficult to determine the percentile concentration of light elements in a sample using the EDS technique. Overlapping of peaks, a result of low spectral resolution, may be counted as an additional disadvantage of EDS. In general, the accuracy of a measurement depends on the mass concentration of heavy elements in the emitting zone whereas the minimal detection limit amounts to 0.1% by weight. Depending on the accelerating voltage and the material type the X-ray emitting zone is in the order of 2–5 μm. This means that the micro-volume of the analyzed specimen is derived only from the subsurface layers of the material. Due to the limitations listed above the chemical composition of the samples tested in the experiment described here was determined independently, in a manner alternative to the classical EDS, and in our opinion yielding more reliable results. The optical emission spectrometer (OES) was applied for this purpose [8].

## 6. Detailed Results of the Tests Conducted and Their Interpretation

### 6.1. Results Obtained on Samples Made of S355J2+N Steel

The *S355J2+N* steel is a typical structural low carbon manganese steel of ferritic-pearlitic microstructure. It is weldable and exhibits good impact strength (even at sub-zero temperature values). The material used for the tests underwent normalization. Detailed metallographic examination showed that the structure of this steel contained elongated inclusions of manganese sulfides [8].

The chemical composition of the tested samples made of *S355J2+N* steel was identified with the Foundry–Master optical emission spectrometer (Worldwide Analytical Systems, Uedem, Nordrhein-Westfalen, Germany) and is listed in the Table 2 (according to [8]).

Figure 10 and Figure 11 depict impact fractures obtained during our research. Figure 10 refers to the tests conducted at +20 °C, while Figure 11 refers to the analogous test conducted at −20 °C. The zones of stable fracture growth and zones of plastic deformation appearing mainly at the side edges of the tested sample, usually referred to as the plastic lips (Figure 7), are indicated on each picture. Each photo of a presented impact fracture is accompanied by a corresponding *F–s* curve with representative values, indicating the fracture mechanism occurring during the experiment (Figure 5a,b). These values should be interpreted as average values estimated on homogeneous six-element test sample. Each representative average value of the random variable *F* (kN) (or random forces *F_gy_* and *F_m_*, respectively) is accompanied by the averaged displacement of the force application point *s* (mm). In addition, empirically estimated coefficients of variation, namely the ν_F_ coefficient—a variation measure of the random variable *F* (measured along the direction parallel to the vertical axis of the *F–s* graph), and the ν_s_ coefficient—a variation measure of the random displacement *s* induced by this force (indicated along the direction parallel to the horizontal axis of this graph) are shown as well. Thus, all the graphs depicted in red are the averaged ones. These graphs fit within the bounds drawn in black and determined as the average value decreased and increased by one standard deviation computed for the random force *F* (estimated on the statistical sample). The ranges in green, drawn at representative values of the force *F* indicated in every picture (along the vertical direction) and accompanying displacement *s* (along the horizontal direction) represent the measure of their random variability determined at the level of a single standard deviation calculated with respect to the proper average value. Sample numbering conforms to the key listed in Table 1.

One may easily notice that in all the cases depicted in Figure 10, the obtained impact fractures exhibit a dominant stable fracture growth zone. This result is confirmed by the accompanying *F–s* curves, which may be assigned to the category *F* (Figure 5b) in each of the considered cases (according to [15]). The force *F_iu_* does not manifest itself clearly on any of the graphs. This means that all the stages of the fracture were completely stable, and therefore self-arresting of uncontrolled micro-fractures growing in the affected material was fully successful. Referring to this steel grade, typical for the material of ferritic-pearlitic structure, the steel cooled down after exposure to the simulated fire incident exhibited slightly worse plastic properties than the same steel unaffected by the action of simulated fire. The *F–s* curve related to the sample denoted by *1 (+20)* (Figure 10a) proved to be significantly more “developed in the horizontal direction” than all the remaining curves depicted in a sequence in Figure 10b–e. This is to be associated with significantly higher energy *W_t_* needed to break the sample, and therefore significantly better impact strength. The macroscopically observed plastic deformation of the sample proved to be much bigger in this case as well, thus confirming the above statement. When referring to the sample heated up to 600 °C, no significant influence of the cooling mode applied on the impact strength obtained during the experiment was observed (Figure 10b,c). However, the ductile delamination manifested itself in the material cooled in the furnace (Figure 10b). This has never been observed in analogous scenario of samples cooled down much more rapidly in water mist (Figure 10c). The reduction in impact strength related to the reference value determined earlier on the sample denoted as *1 (+20)* (Figure 10a) and observed on the samples heated at 800 °C (Figure 10d,e) proved to be significantly smaller than the one observed on the samples heated at 600 °C only and depicted in Figure 10b,c. This is a beneficial influence of structural changes generated at such high temperature and associated with austenitic transformation. However, this happened only when the sample after heating was cooled in the furnace (Figure 10d). Rapid cooling in the water mist did not yield such a difference (Figure 10e), as it resulted in local hardening of the material. This in turn induced higher susceptibility to brittle fracture. These conclusions seem to be in complete agreement with results of earlier research conducted by the authors, reported in the papers [6,7,8].

The impact fractures obtained on samples made of *S355J2+N* steel tested at −20 °C and juxtaposed in Figure 11 exhibit completely different characteristics.

The stable fracture growth zone observed in this testing scenario is very limited in every case. An unstable fracture area with adjoining plastic zone dominates the picture this time. Such morphology of obtained fracture surfaces is consistent with accompanying *F–s* graphs, as one may clearly identify the *F_in_* force on each of these. After the value *F_in_* is reached, the transferred force is abruptly reduced at minimal increase of the accompanying displacement *s*. All the *F–s* graphs presented in Figure 11 should be assigned to category *D* according to the code [15]. In the test scenario considered, simulating the winter conditions, the impact strength of the tested steel is significantly reduced and potential self-arresting of the micro-cracks generated in the material is not very effective. Interestingly, in this case, surviving a fire incident (Figure 11b–e) followed by cooling proves to be beneficial, as it results in significant increase in impact toughness when compared against the sample denoted *1 (−20)* (Figure 11a). This increase was the highest when the sample was heated to the temperature of 800 °C and this was followed by slow cooling down of the sample in the furnace (Figure 11d). Under such conditions, the temperature acting on the steel was sufficiently high to induce in the material changes in its microstructure beneficial from the point of view of impact strength [8]. Importantly, this phenomenon has never been negated by local hardening through rapid cooling of the steel in water mist.

Values of the parameter *SFA*_(*n*)_ (%) calculated for the *S355J2+N* steel are listed in the Table 3. Formulas (1)–(4) were used to determine the values listed for *n =* 1, ..., 4, while for *n =* 5, a direct approach depicted in Figure 8 had been used.

These results quantitatively confirm the conclusions drawn in this paper. It has to be noted here that the *SFA*_(1)_ parameter proved to be the most conservative one in this listing, while the *SFA*_(4)_ parameter yielded the least conservative results. The approach based on the *SFA*_(5)_ parameter usually yielded intermediate estimates.

The measured and later averaged values of the lateral expansion parameter *LE* (mm) are juxtaposed below. Figure 12 depicts appropriate average values and coefficients of variation estimated on the statistical sample for each of the samples and test scenarios executed in practice. The presented results confirm the conclusions drawn above.

Figure 13 depicts the values *LE* presented previously in Figure 12 in relation to the corresponding values of energy *W_t_* (J) required to fracture the considered sample. This picture indicates that change in the test temperature from +20 °C to −20 °C results in significant decrease in this energy. This means that under those conditions, the impact toughness of given material is much lower.

### 6.2. Results Obtained on Samples Made of X20Cr13 Steel

This steel is a single phase stainless high alloy chromium steel, representative for a group of special steels resistant to abrasion. The microstructure of the steel of this type is purely martensitic. This material, characterized by good resistance to corrosion in moderately aggressive environments lacking chlorine content, is usually applied in the tempered and annealed state. It has to be heated to 300–400 °C prior to welding, and after welding it has to be annealed to soften.

The chemical composition of the samples made of this steel, identified using OES [8], is listed in Table 4.

The impact strength fractures obtained on samples made of this steel are depicted in Figure 14 for the tests conducted at +20 °C, and in Figure 15 for the tests conducted at −20 °C. Each of the presented fracture surfaces is accompanied by a corresponding *F–s* curve. Numbering of the samples conforms to the key listed in Table 1.

One may easily notice that in each of the considered testing scenarios, a brittle fracture was obtained, meaning that the failure of this steel under applied external load would be an abrupt phenomenon, without preceding signs of increasing weakening. Interestingly, the same conclusion may be drawn regarding the sample denoted as *2 (+20)*, which was not affected by prior action of fire temperature (Figure 14a). This statement is confirmed by the *F–s* curves corresponding to respective fractures, as in each of the considered testing scenarios, the limiting force *F_in_* reveals itself, initiating the unstable fracture growth phase in the material. Thus, self-arresting of these fractures proved to be ineffective in this case. In addition, on all of the *F–s* graphs presented here, the equivalence *F_m_ = F_in_* holds. Therefore, all these cases may be assigned to category *D* (according to [15]). The results obtained here, due to the brittle behavior of the material revealed during the tests, in general disqualify any possibility for future application of this steel in construction. This brittleness is a result of material hardening by the martensitic structure intentionally induced in it during manufacture.

The results depicted in Figure 15, pertaining to the tests conducted at −20 °C not only confirm but even clearly reinforce the above statement, formulated with respect to analogous samples made of the same steel but conducted at +20 °C. The effective self-arresting of fractures, initiated when the force *F_a_* appears on the *F–s* graph (Figure 5a), under such conditions becomes negligibly small.

The conclusions drawn above are formally confirmed at first in the values of *SFA*_(*n*)_ parameters measured during experiment and juxtaposed in the Table 5 as well as in Figure 16 and Figure 17 depicting the values of the *LE* parameter obtained during each considered testing scenario and its relation to the energy *W_t_* needed to break the sample.

In the case of this steel, should one assume the values quantified by the parameter *SFA*_(5)_ to be authoritative, then all the remaining parameters beginning with *SFA*_(1)_ and ending with *SFA*_(4)_ clearly overestimate this value.

Negative values of the *LE* parameter obtained for this steel during the tests conducted at −20 °C, accompanied by very low impact strength determined, should be understood as a proof of lateral contraction (*LC*), depicted in Figure 6, occurring in the analyzed cross section.

The data depicted in Figure 15 indicate one more interesting fact. The simulated fire action episodes in the case of tests conducted at +20 °C resulted in a significant reduction in the already very low impact strength of the material. This reduction was slightly smaller when the considered steel was previously heated up to 800 °C, meaning that permanent changes occurring in the microstructure of the material were beneficial from the point of view of this property. Rapid cooling of the material in water mist almost completely eliminated this effect. However, when the tests were conducted at −20 °C (simulating winter conditions), then the preceding episodes of simulated fire action proved to be in general beneficial to the material. This effect was somewhat diminished when the samples were heated up to 800 °C. Nevertheless, the impact strength of this steel related to the testing conditions was clearly negligibly small.

### 6.3. Results Obtained on Samples Made of X6CrNiTi18-10 Steel

This is an acid resistant, nonmagnetic steel, with low yield limit (of about 220 MPa). It exhibits good impact strength even at sub-zero temperature and good mechanical properties when exposed to fire. Furthermore, it exhibits good weldability but simultaneously poor susceptibility to mechanical and electrochemical polishing. The microstructure of such steel consists of austenitic matrix with small amount of titanium carbide precipitates.

The chemical composition of the samples made of this steel identified by the OES spectrometer [8] is listed in Table 6.

The impact strength test fractures obtained on samples made of this steel accompanied by corresponding *F–s* curves are depicted on Figure 18 for the tests conducted at +20 °C, and on Figure 19 for the tests conducted at −20 °C.

All the fractures presented in this group exhibited substantial permanent plastic deformations. This indicates a completely plastic character. The morphology of the examined fractures, in each of the considered scenarios, is dominated by a zone of stable fracture growth. This conclusion is supported by the shape of the *F–s* curve accompanying each case. These curves, according to the classification contained in [15] may be assigned to category *F* (Figure 5b). A large area bounded from above by the *F–s* curve is a measure of adequately high impact strength. It is to be correlated with a high capability to effectively self-arrest micro-cracks induced in the material.

Observation of the impact test fractures obtained during the tests made at −20 °C (Figure 19) leads to analogous conclusions. However, under those conditions, impact strength was proven to be slightly lower. Nevertheless, even in those cases, the obtained *F–s* curves may be assigned to the category *F* of the classification presented in the code [15] (Figure 5b). The ductile delaminations of the material, occurring during the tests conducted in such conditions and indicating its slightly lower resistance to the stresses accumulating in the sample as a result of dynamic loads applied [39], seem to represent a certain qualitative difference here.

The estimates of the *SFA*_(*n*)_ (%) parameters conducted by us yielded, regardless of the measurement method applied, unequivocal confirmation of the fully plastic character of the fractures obtained. These parameters are juxtaposed in Table 7.

The measurements of the *LE* parameter showed a seemingly unexpected effect, that for the considered steel plastic deformations on the impact strength test, fractures were quantitatively larger on the samples tested at −20 °C (Figure 20). This was clearly confirmed by the relation presented in Figure 21.

This relation (Figure 21) in the case of considered steel grade exhibits peculiar characteristics. On one side it shows very high post-fire impact strength of the tested material, a factor not surprising in the case of steels exhibiting austenitic structure. However, this impact strength decreases significantly when tested at simulated winter conditions, nevertheless remaining at sufficiently high level. On the other side, contrary to the results obtained and presented in this paper for other steel grades, the results of tests conducted at −20 °C are located to the right of the corresponding results obtained at +20 °C. Moreover, for this steel, the fire incident simulated at 600 °C proved to be beneficial, as it raised the impact strength tested after cooling, regardless of the cooling mode applied. The higher temperature acting on the steel (800 °C), allowing for the structural changes to occur, finally proved to be detrimental to the tested material (when compared against the test made on samples denoted as *3 (+20)*, as well as *3 (−20)—*Figure 18a and Figure 19a).

### 6.4. Results Obtained on Samples Made of X2CrNiMoN22-5-3 Steel

The steel of this grade is classified as a typical stainless and acid resistant high alloy steel which is resistant to pitting and also to surface corrosion. It is characterized by two-phase, austenitic-ferritic, chromium–nickel–molybdenum microstructure of duplex type. This steel is recommended for application at temperatures below 300 °C due to the occurrence of the detrimental 475 °C brittleness phenomenon.

The chemical composition of the samples made of this steel identified by the OES spectrometer [8] is listed in the Table 8.

The impact strength test fractures obtained on samples made of this steel grade are depicted in Figure 22 for the tests conducted at +20 °C and in Figure 23 for the analogous tests conducted at −20 °C. In all the considered cases, presented fracture surfaces are accompanied by a corresponding *F–s* curve.

The impact strength fractures obtained on samples made of this steel in +20 °C exhibited completely ductile character. The macroscopically observed changes in shape testify to that. The important qualitative difference with respect to the analogous testing scenarios executed on samples made of *X6CrNiTi18-10* steel appear as multiple transverse delaminations. So far, these delaminations occurred only in samples tested in −20 °C. This destruction mode at completely plastic fracture is usually explained as a result of local heterogeneities in plastic properties of tested material [39]. This phenomenon often causes an incomplete fracture during classical impact strength tests, ending in following quasi static pushing of the sample through hammer clamps. These statements seem to be confirmed by the *F–s* graphs accompanying these fractures, as one may easily identify a horizontal plateau generated during the finishing stage of fracturing the samples (Figure 22a–c). All the *F–s* graphs presented in Figure 22 clearly may be assigned to the category *F* (according to [15]).

This steel exhibited a peculiar reaction to the simulated fire episode. This was especially true when the steel was heated to 800 °C. This temperature level resulted in changes in the microstructure of the affected samples. Under those conditions, regardless of the cooling mode applied, the impact strength test fractures were accompanied by very small displacements *s*, indicating significant deleterious changes in the impact strength of the tested steel. The complementary metallographic analysis conducted by us showed that heating of the steel up to 800 °C resulted in intensive precipitation of numerous carbides, nitrides, and intermetallic phases out of supersaturated ferrite and austenite in its microstructure [8]. Additionally, under such conditions slow cooling of the sample in the furnace (the sample denoted as *480 (+20)*—Figure 22d) yielded longer material transition time through the so-called 475 °C brittle zone. During this phase, usually deleterious brittle phases appear in the material, additionally weakening it, and this in turn obviously decreases its impact strength. Fast cooling in water mist (the sample denoted as *481 (+20)*—Figure 22e) resulted in shorter transition time through this zone. As a result, the material weakening proved to be less significant in this case [8].

Analogous development was observed in the case of samples made of this steel grade and heated up to 600 °C only. During slow cooling in the furnace (the sample denoted as *460 (+20)*—Figure 22b), the transition time through the 475 °C brittle zone proved to be sufficiently long to significantly decrease the post-fire impact strength of the tested steel (as compared against analogous steel not subjected to the simulated fire action—Figure 22a). However, this phenomenon did not appear at all when the samples were cooled sufficiently fast (Figure 22b).

The fractures observed during the tests conducted at −20 °C (Figure 23) proved to be fully plastic as well, exhibiting properties analogous to those described above with respect to the tests conducted at +20 °C. Interestingly, under those conditions, no significant deterioration of the impact strength revealed was observed. Though quantitative differences were observed, their formal importance seems to be negligible. In all considered scenarios, the *F–s* graphs were assigned to the category *F* (according to [15]). This conclusion is clearly confirmed by the juxtaposition presented in Table 9.

Let us note that, as shown in the Figure 24, in the case of this steel grade in all the testing scenarios considered, the values of the *LE* parameter obtained during the tests conducted at +20 °C happened to be higher than analogous values obtained during the tests conducted at −20 °C. However, this does not mean that the impact strength revealed during those tests proved to be better in each of the compared scenarios. A comparison of the samples denoted as *4 (+20)* and *461 (+20)* (Figure 25) indicates a not-so-obvious statement that lowering the test temperature to −20 °C resulted in increased impact strength.

## 7. Concluding Remarks

In the authors’ opinion, the results presented in this paper confirm the need for detailed analysis of impact strength test fractures obtained experimentally in order to reliably draw conclusions on the possible future safe application of material which has survived a fire incident. In particular, one has to determine whether the material affected by high temperature during fire has retained the capability to safely resist the external loads applied to it with sufficiently high probability. The considered fractures were analyzed a posteriori, i.e., after finished simulated fire incidents, on the samples made of a given steel grade cooled down successfully after prior exposure to high temperature. In that sense, the quantitative and qualitative estimation of given steel grade post-fire impact strength is an action of fundamental importance preceding the decision whether and to what extent the material may remain in continued service after fire. Therefore, it is postulated to include this test in the engineering practice of traditionally performed visual inspection of the deformed structure performed on site, accompanied by experimental verification of the basic mechanical properties of affected material. Obtained results should warrant that during prolonged service after a fire, no brittle fractures would occur in bearing steel structural components, as these fractures during unstable development may result in structural failure. Sufficiently high impact strength of the material warrants that the possible brittle fractures will be effectively suppressed.

The reliability of the conclusions drawn based on the approach described in this paper is enhanced by the fact that the results obtained are confirmed by several methods applied. Regardless of the testing scenario applied to check a given steel grade, similar results were arrived at for *LE* parameter, the *LE–W_t_* relationship, the *F–s* curve, or only a simple observation of the morphology of experimentally obtained fracture surface.

It was proven that various steel grades behave differently after surviving a fire incident, in a manner that is not obvious and difficult to foresee if not supported by appropriate specialized tests. Steel grades varying in intentionally induced microstructure have been selected for analysis, as qualitatively different response of tested materials to fire exposure occurring under the same controlled conditions was expected. The a priori formulated thesis on the differences in the impact strength exhibited post-fire by various steel grades under different thermal conditions was experimentally confirmed. Therefore, should a prospective user expect a practically relevant and useful information on the impact strength of tested steel, this information has to be accompanied by the knowledge whether it concerns winter or summer service conditions of affected facility. The differences in this aspect, identified depending on such conditions, may prove to be very significant.

The tests conducted by us have shown that for some steel grades considered in this paper, lowering the test temperature from +20 °C to −20 °C resulted in improved impact strength of the material. Previous fire incidents did not always weaken the tested material as well. In many computational cases important in practice, especially those related to simulation of winter service conditions, the impact strength determined post-fire proved to be higher than that exhibited by the same material not exposed to fire action. However, under those circumstances in general, the final conclusions proved to be the opposite, when the impact strength tests were conducted in a manner simulating summer conditions (that means at +20 °C).

The results obtained were strongly affected by the mode applied to cool the hot steel. In most cases, rapid cooling of samples in water mist, simulating an action of firefighters, resulted in material hardening. This was equivalent to lowering its impact strength. However, this conclusion was not confirmed in the case of a duplex steel, as for this steel, fast reduction of temperature meant fast transition through the deleterious 475 °C brittle zone, and therefore yielded lower degree of material degradation in effect.

The key influence of temperature level reached during the considered fire incident on post-fire properties of analyzed material, and especially its impact strength, was confirmed by this research. Reaching a temperature level initiating structural changes in the considered steel seems to be the key here, as these changes are usually accompanied by developments associated with rebuilding of crystalline lattice and often with numerous precipitates of weakening character.

Continuing this research, we intend to find out how the material properties verified after a fire incident are affected by the extended heating time. It seems that the prolonged effective fire exposure occurring at the same temperature level should in practice result in additional structural changes significantly affecting the post-fire durability of the analyzed steel. These changes occurred so slowly that they did not have sufficient time to reveal themselves during the study scenario analyzed in this article and associated with one-hour heating time.

We also intend to verify how the change in heating speed while keeping the same total heating time affects the post-fire properties of given steel grade. It seems, in this context, that steel heated at a lower temperature increase rate may exhibit a better capacity to accommodate a higher temperature at the same safety level.

## Figures and Tables

**Figure 1 materials-16-03281-f001:**
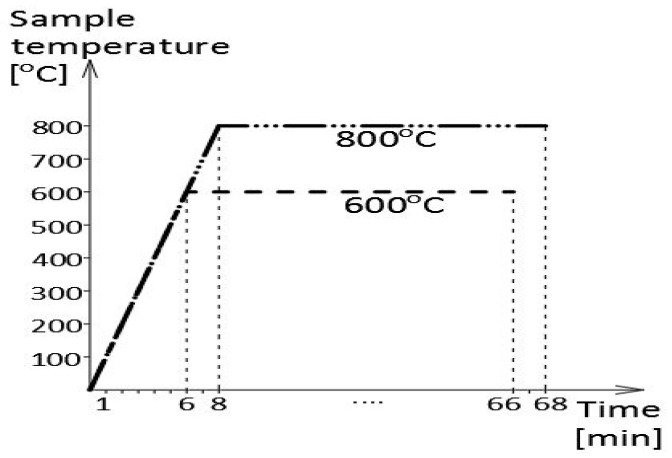
The initial thermal treatment of samples simulating the fire action (according to [7]).

**Figure 2 materials-16-03281-f002:**
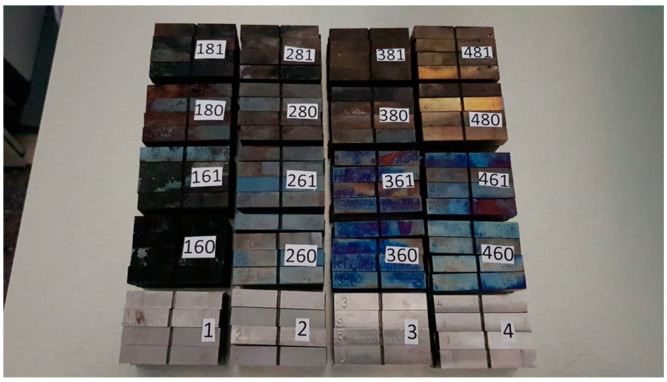
Samples prepared for impact strength testing, made of *S355J2+N* steel (1st set), *X20Cr13* steel (2nd set), *X6CrNiTi18-10* steel (3rd set), and *X2CrNiMoN22-5-3* steel (4th set), from left to right.

**Figure 3 materials-16-03281-f003:**
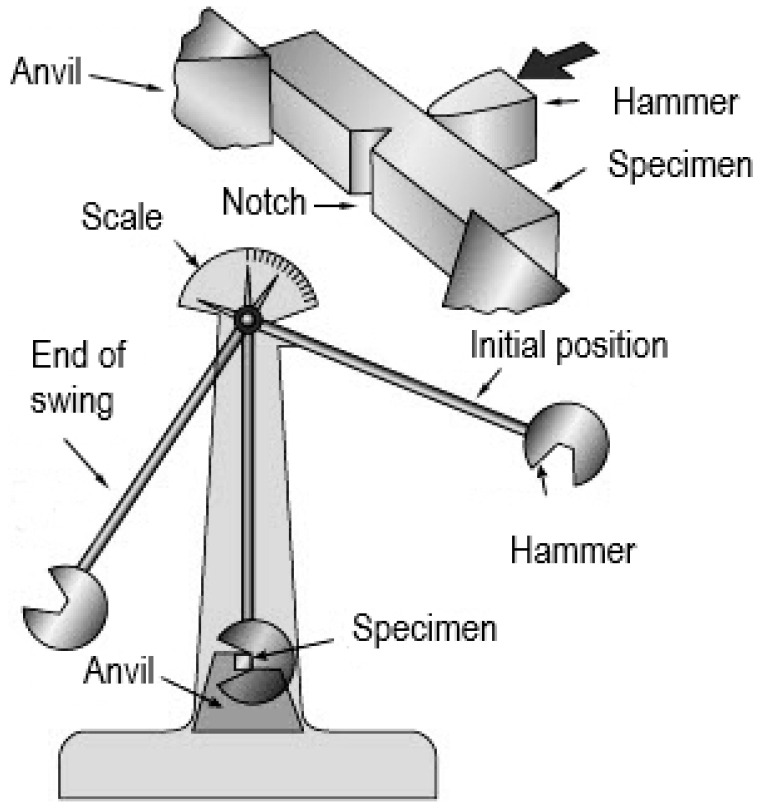
Principle of executing Charpy impact strength test.

**Figure 4 materials-16-03281-f004:**
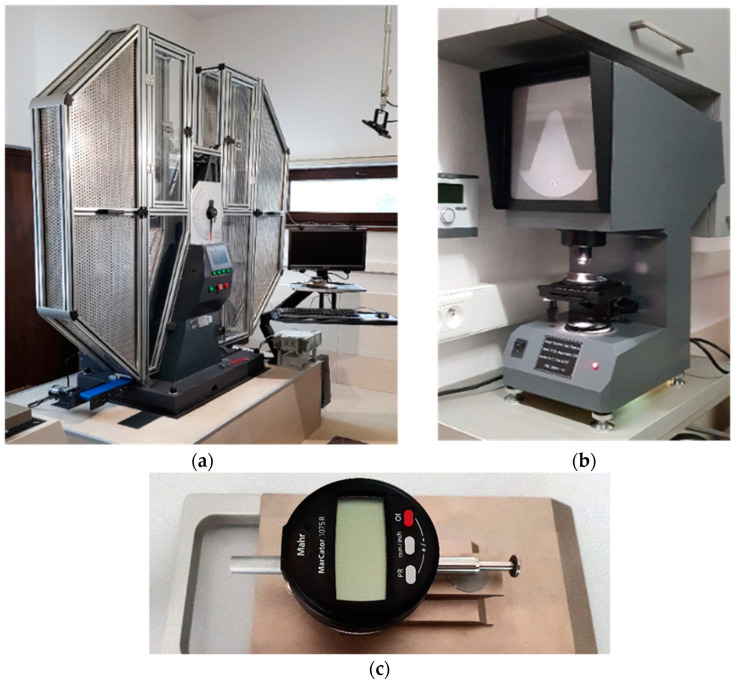
Testing equipment used during research: (**a**) instrumented Charpy hammer of the JB-W450E-L type, (**b**) XT-50 notch profile projector, (**c**) the gauge used to measure the lateral expansion of a sample (according to [7]).

**Figure 5 materials-16-03281-f005:**
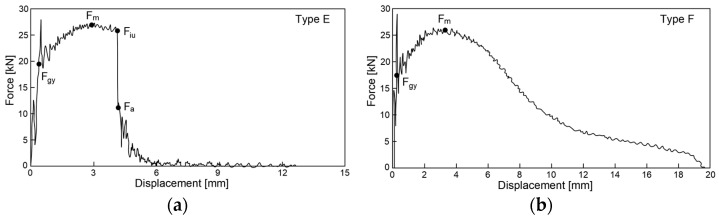
The force–displacement of force application point graphs as characteristics of typical impact strength fractures including: (**a**) category *E* fracture type, (**b**) category *F* fracture type. Qualification following the recommendations of [15].

**Figure 6 materials-16-03281-f006:**
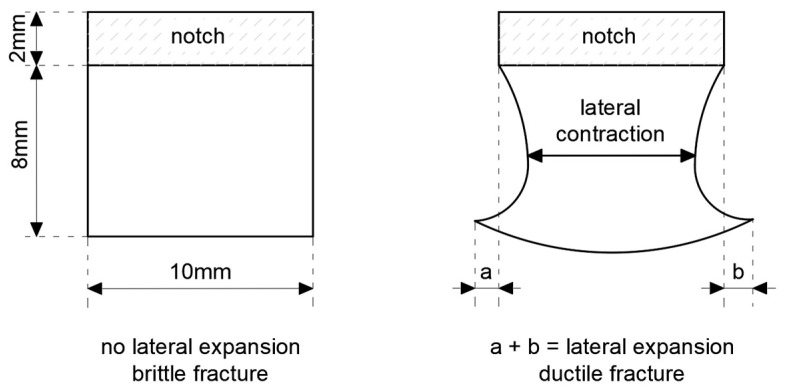
The *LE* (mm) lateral expansion parameter measurement concept.

**Figure 7 materials-16-03281-f007:**
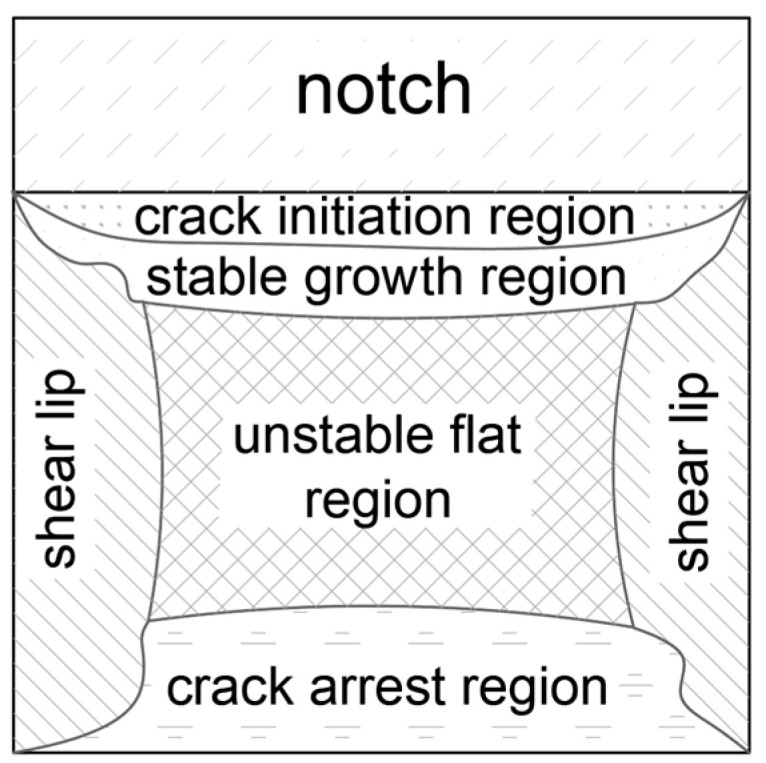
Fracture development stages in an ISO Charpy V sample on the typical fracture surface of mixed ductile character.

**Figure 8 materials-16-03281-f008:**
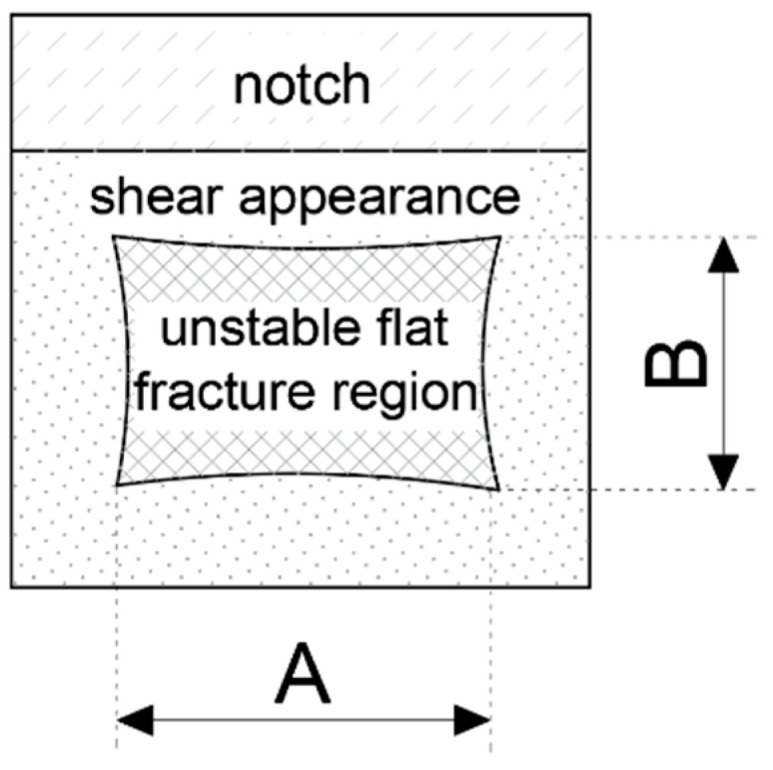
Flat fracture region with partially cleavage cracked zone (partially brittle) and surrounding shear generated fracture area.

**Figure 9 materials-16-03281-f009:**
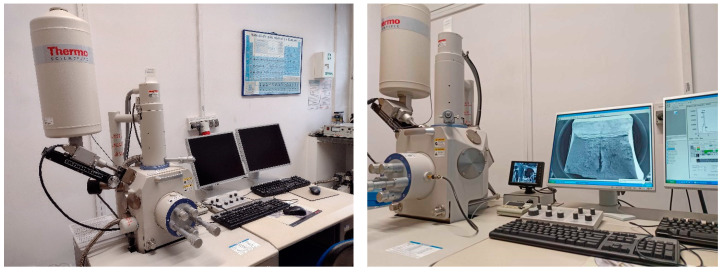
Scanning electron microscope (SEM) used to analyze the microstructure of considered steel.

**Figure 10 materials-16-03281-f010:**
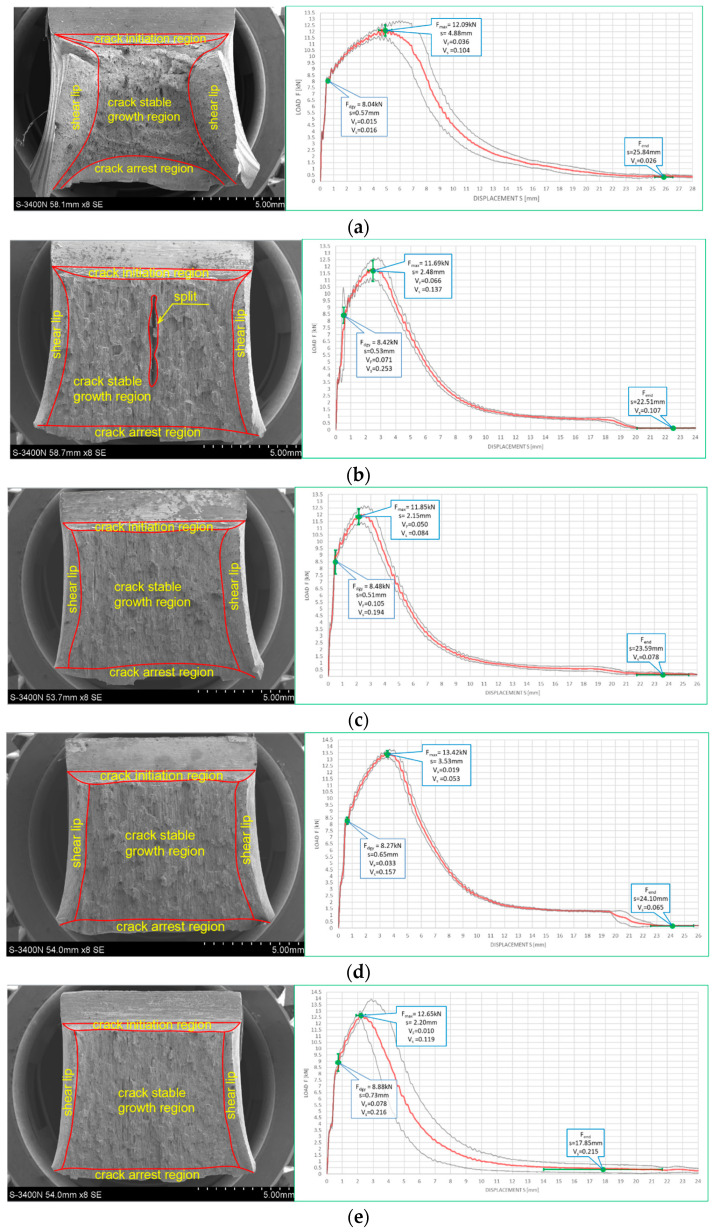
Morphology of the impact fractures obtained during the tests conducted at +20 °C on the samples made of *S355J2+N* steel—denotations follow the key listed in Table 1. (**a**) *1 (+20)*; (**b**) *160 (+20)*; (**c**) *161 (+20)*; (**d**) *180 (+20)*; (**e**) *181 (+20)*.

**Figure 11 materials-16-03281-f011:**
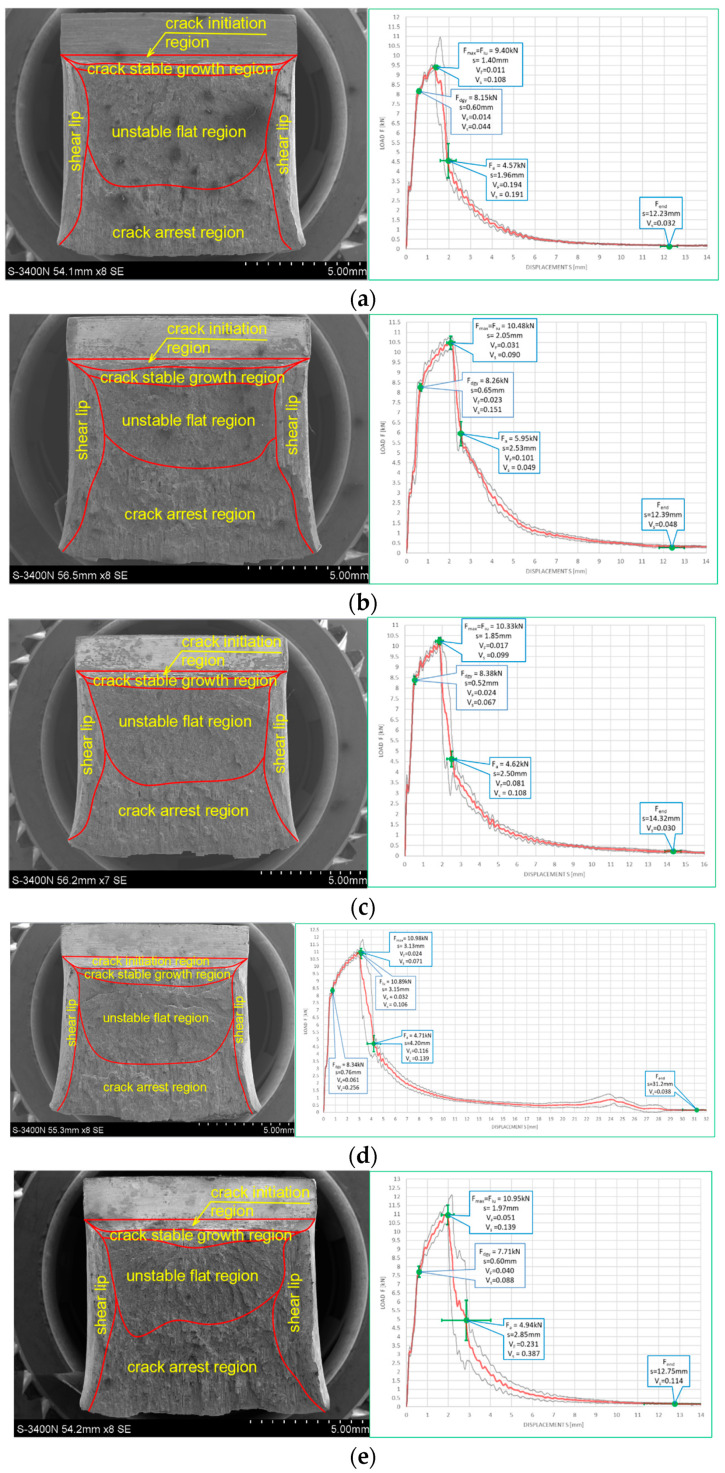
Morphology of the impact fractures obtained during the tests conducted at −20 °C on the samples made of *S355J2+N* steel—denotations follow the key listed in Table 1. (**a**) *1 (−20)*; (**b**) *160 (−20)*; (**c**) *161 (−20);* (**d**) *180 (−20)*; (**e**) *181 (−20)*.

**Figure 12 materials-16-03281-f012:**
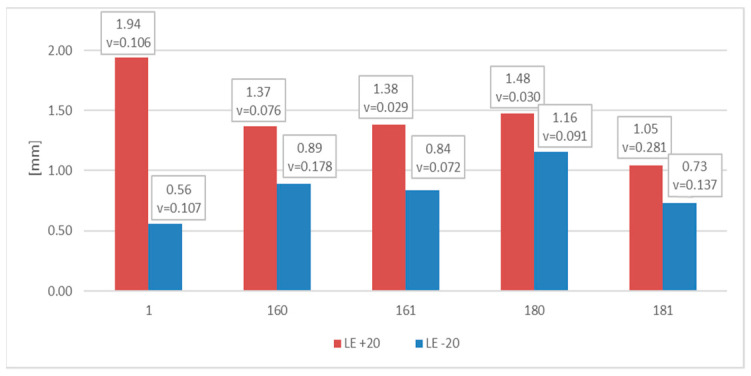
Averaged values of the *LE* parameter and coefficients of variation corresponding to them (estimated on a statistical sample) obtained on impact toughness test samples made of *S355J2+N* steel.

**Figure 13 materials-16-03281-f013:**
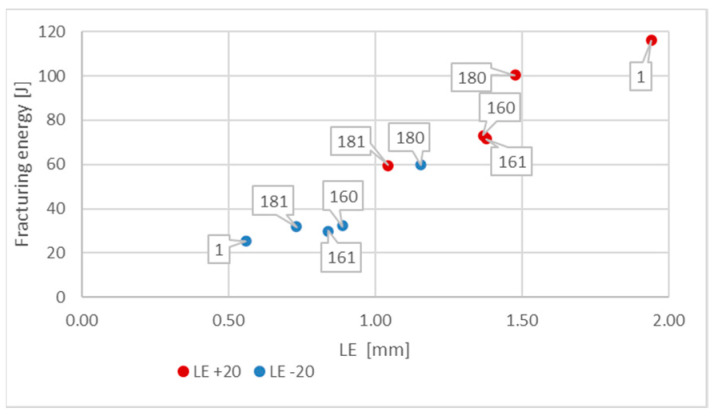
Relationship between averaged values of *LE* parameter and related values of fracturing energy *W_t_* obtained for *S355J2+N* steel at different testing conditions.

**Figure 14 materials-16-03281-f014:**
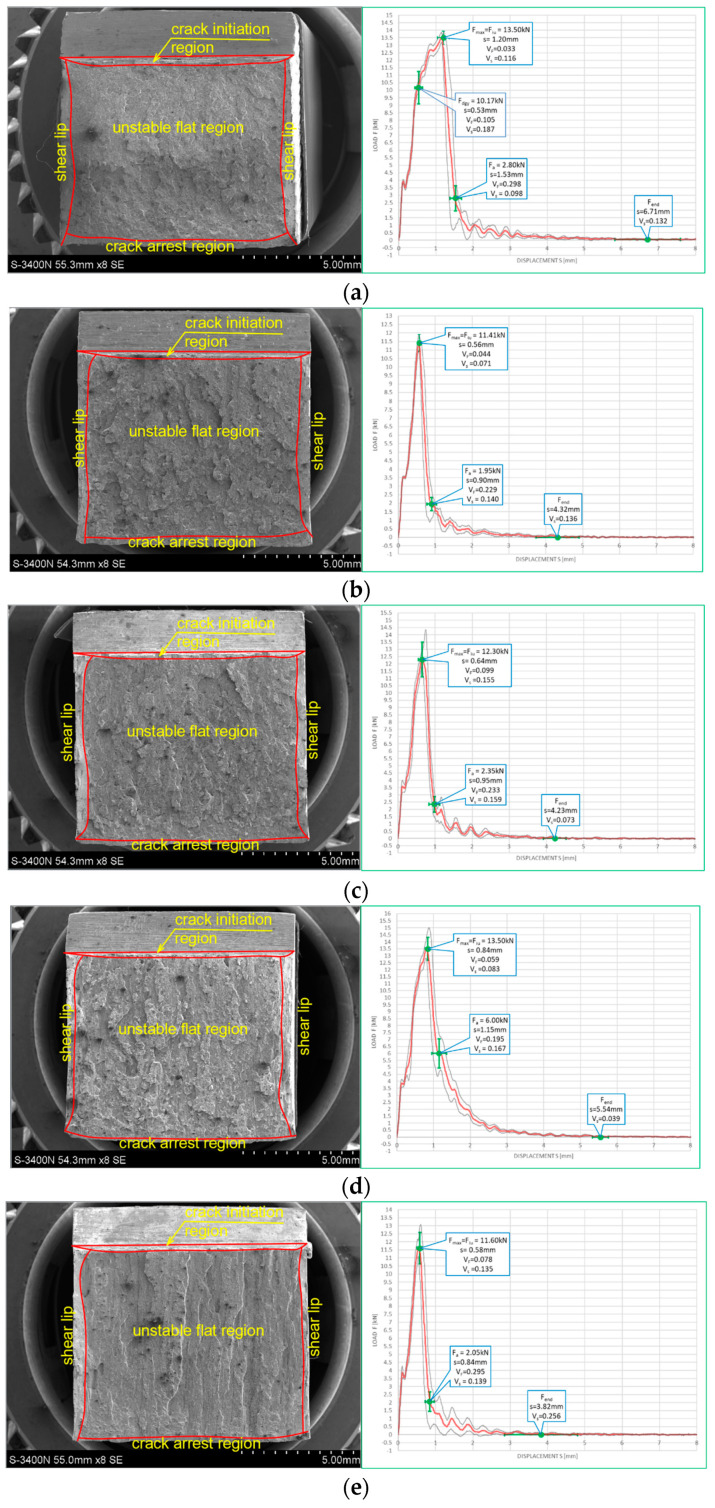
Morphology of the impact fractures obtained during the tests conducted at +20 °C on the samples made of *X20Cr13* steel—denotations follow the key listed in Table 1. (**a**) *2 (+20)*; (**b**) *260 (+20)*; (**c**) *261 (+20)*; (**d**) *280 (+20)*; (**e**) *281 (+20)*.

**Figure 15 materials-16-03281-f015:**
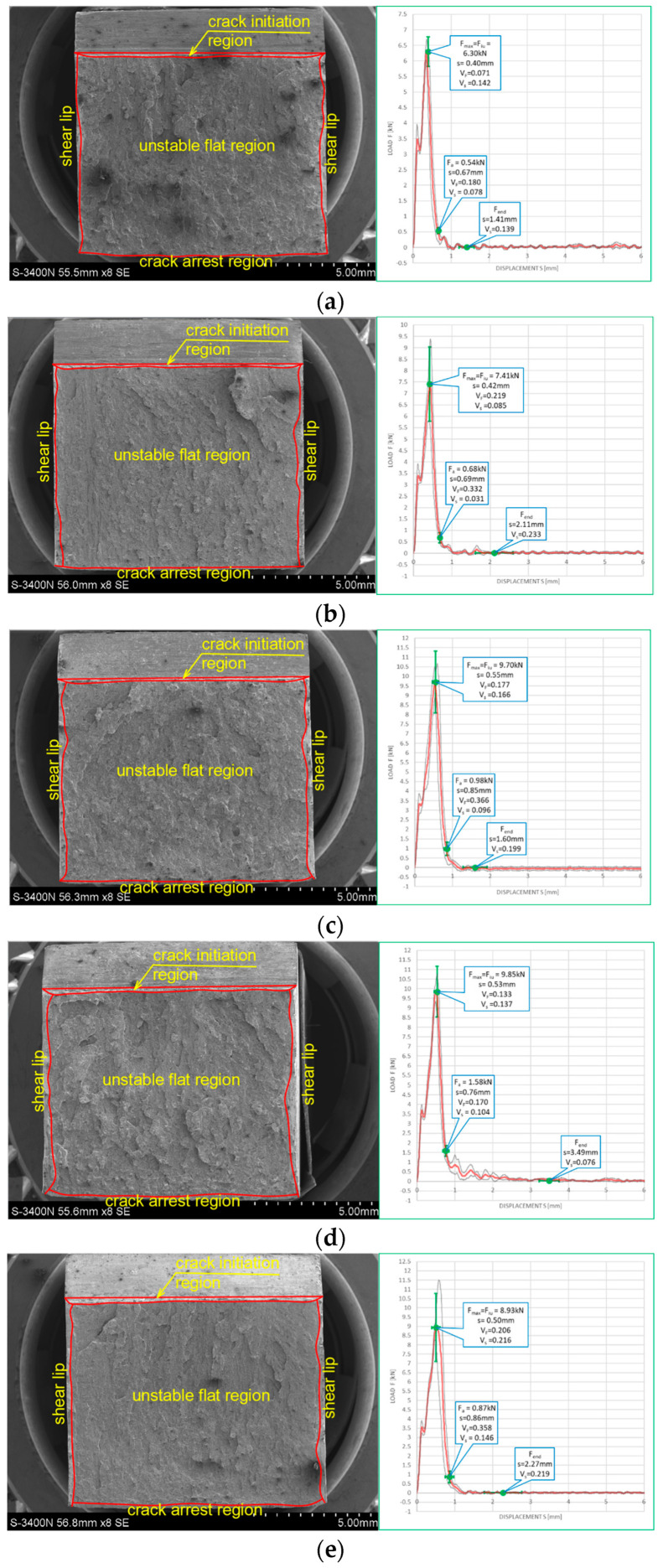
Morphology of the impact fractures obtained during the tests conducted at −20 °C on the samples made of *X20Cr13* steel—denotations follow the key listed in Table 1. (**a**) *2 (−20)*; (**b**) *260 (−20)*; (**c**) *261 (−20)*; (**d**) *280 (−20)*; (**e**) *281 (−20)*.

**Figure 16 materials-16-03281-f016:**
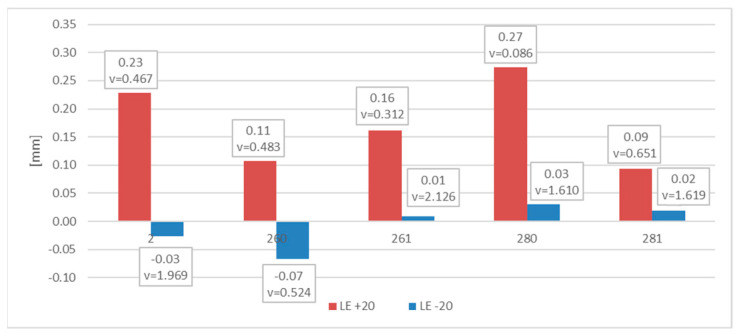
Averaged values of the *LE* parameter and corresponding coefficients of variation (estimated on a statistical sample) obtained on impact toughness test samples made of *X20Cr13* steel.

**Figure 17 materials-16-03281-f017:**
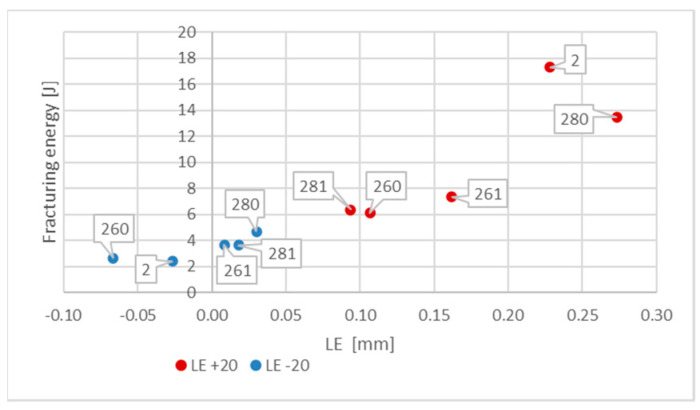
Relationship between averaged values of *LE* parameter and related values of fracturing energy *W_t_* obtained for *X20Cr13* steel at different testing conditions.

**Figure 18 materials-16-03281-f018:**
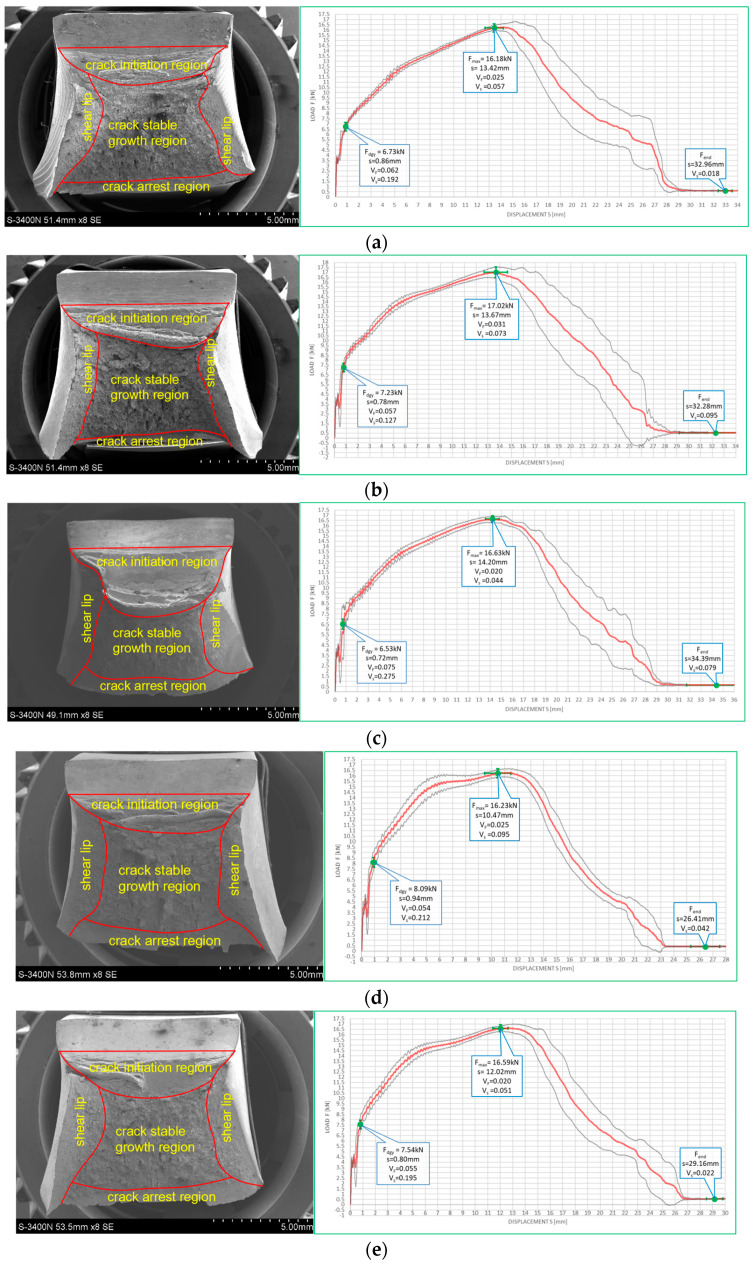
Morphology of the impact fractures obtained during the tests conducted at +20 °C on the samples made of *X6CrNiTi18-10* steel—denotations follow the key listed in Table 1. (**a**) *3 (+20)*; (**b**) *360 (+20)*; (**c**) *361 (+20)*; (**d**) *380 (+20)*; (**e**) *381 (+20)*.

**Figure 19 materials-16-03281-f019:**
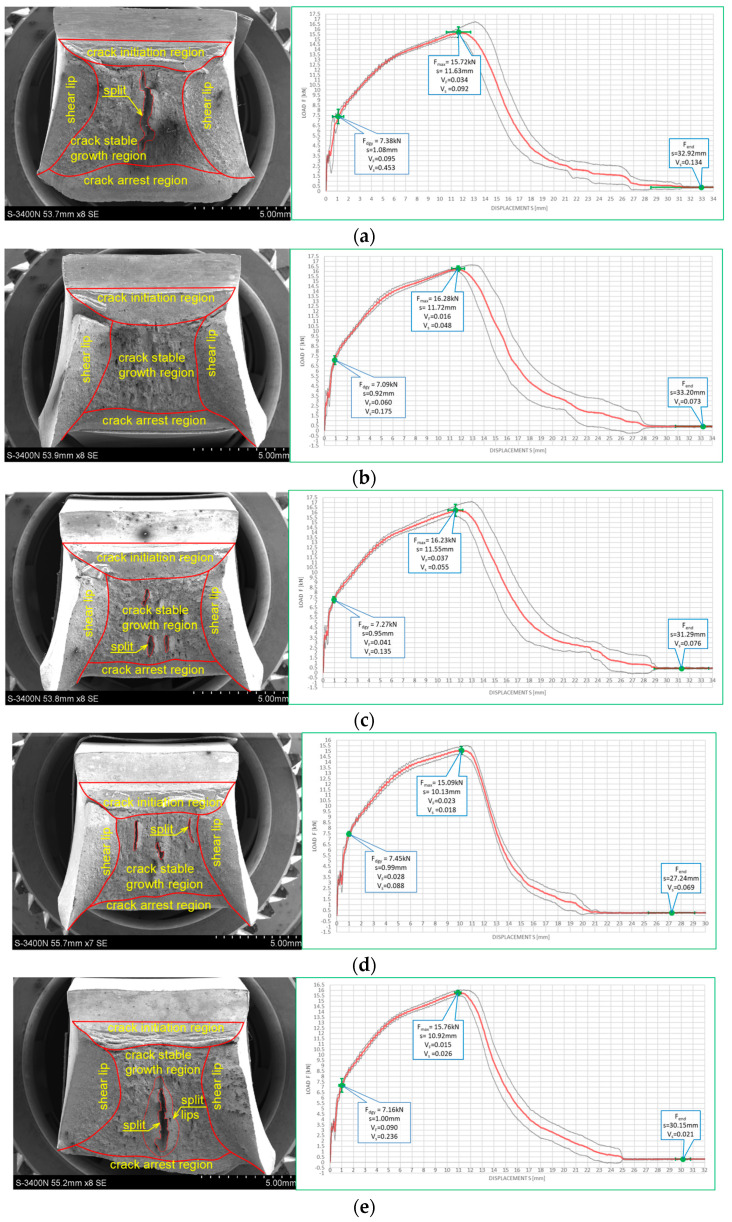
Morphology of the impact fractures obtained during the tests conducted at −20 °C on the samples made of *X6CrNiTi18-10* steel—denotations follow the key listed in Table 1. (**a**) *2 (−20)*; (**b**) *260 (−20)*; (**c**) *261 (−20)*; (**d**) *280 (−20)*; (**e**) *281 (−20)*.

**Figure 20 materials-16-03281-f020:**
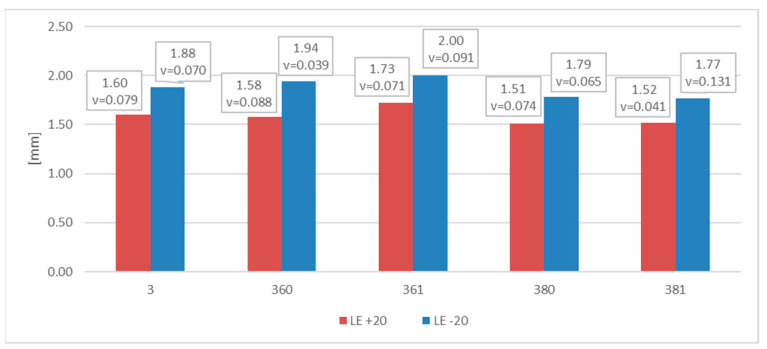
Averaged values of the *LE* parameter and corresponding coefficients of variation (estimated on a statistical sample) obtained on impact toughness test samples made of *X6CrNiTi18-10* steel.

**Figure 21 materials-16-03281-f021:**
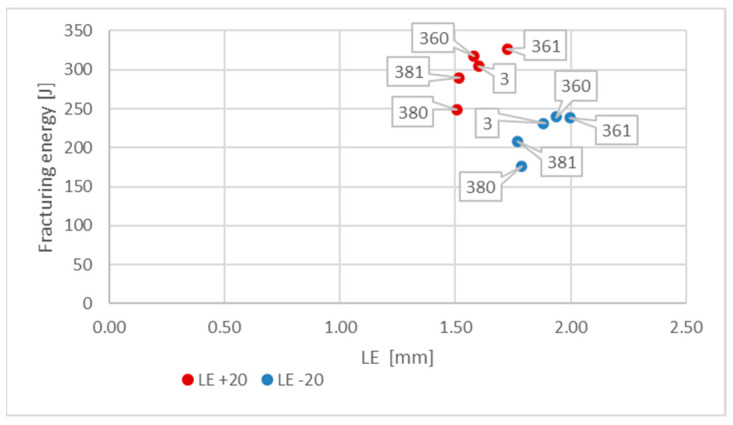
Relationship between averaged values of *LE* parameter and related values of fracturing energy obtained for *X6CrNiTi18-10* steel at different testing conditions.

**Figure 22 materials-16-03281-f022:**
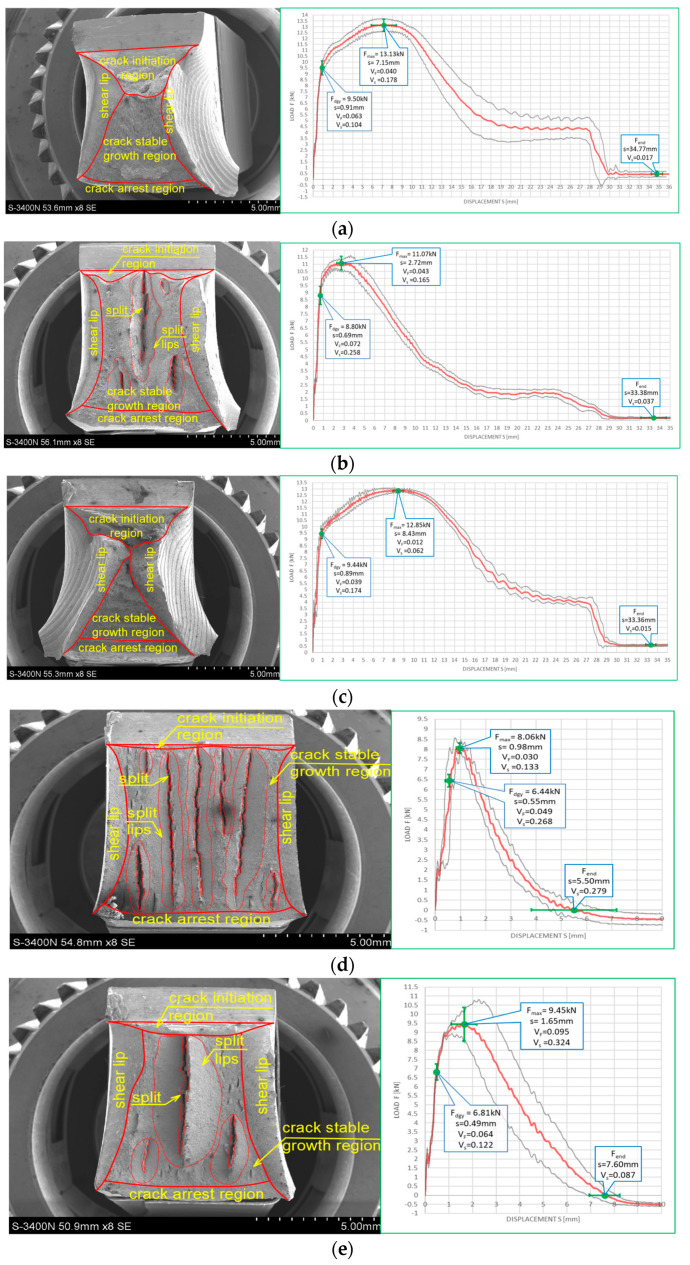
Morphology of the impact fractures obtained during the tests conducted at +20 °C on the samples made of *X2CrNiMoN22-5-3* steel—denotations follow the key listed in Table 1. (**a**) *4 (+20)*; (**b**) *460 (+20)*; (**c**) *461 (+20)*; (**d**) *480 (+20)*; (**e**) *481 (+20)*.

**Figure 23 materials-16-03281-f023:**
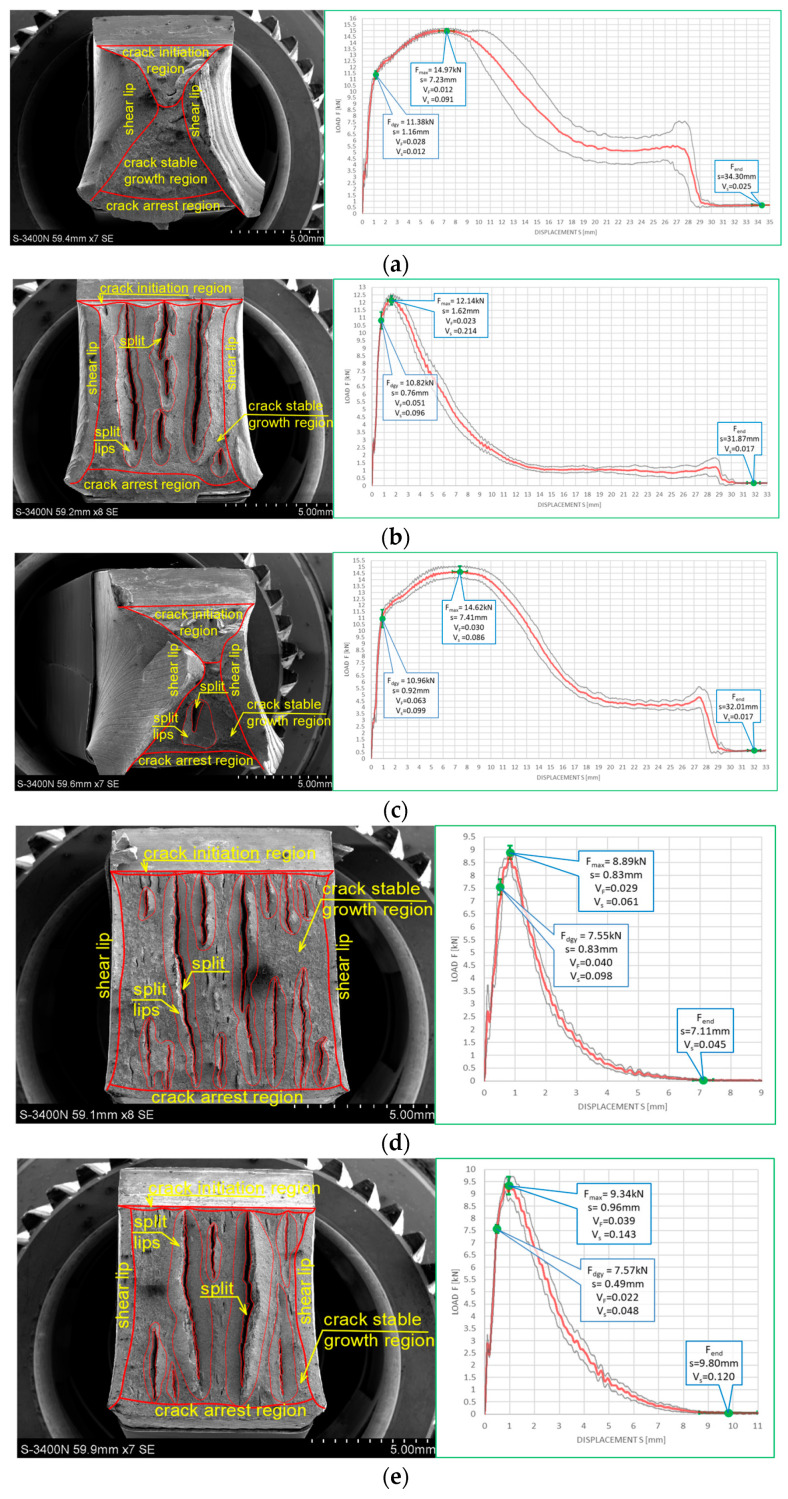
Morphology of the impact fractures obtained during the tests conducted at −20 °C on the samples made of *X2CrNiMoN22-5-3* steel—denotations follow the key listed in Table 1. (**a**) *4 (−20)*; (**b**) *460 (−20)*; (**c**) *461 (−20)*; (**d**) *480 (−20)*; (**e**) *481 (−20)*.

**Figure 24 materials-16-03281-f024:**
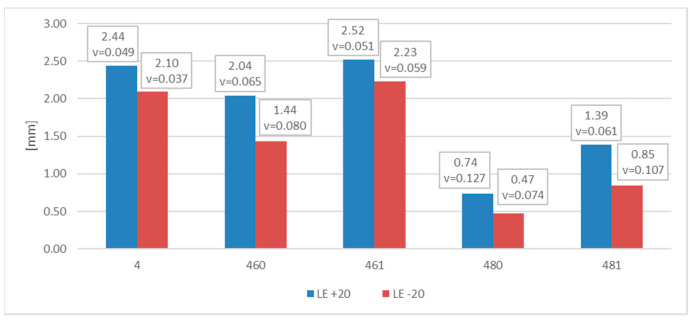
Averaged values of the *LE* parameter and corresponding coefficients of variation (estimated on a statistical sample) obtained on impact toughness test samples made of *X2CrNiMoN22-5-3* steel.

**Figure 25 materials-16-03281-f025:**
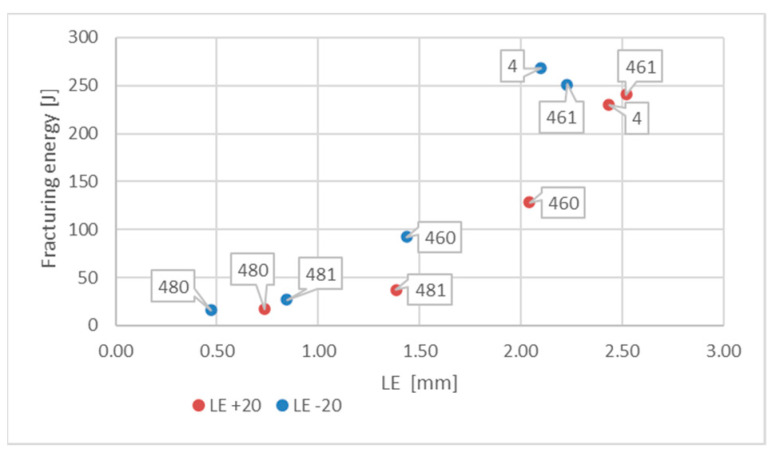
Relationship between averaged values of *LE* parameter and related values of fracturing energy obtained for *X2CrNiMoN22-5-3* steel at different testing conditions.

**Table 1 materials-16-03281-t001:** Description mode of the samples subjected to impact strength tests.

First Digit—Steel Grade	Second Digit—Heating Temperature	Third Digit—Cooling Mode	Additional Information—Testing Temperature
**1**—*S355J2+N* **2**—*X20Cr13* **3**—*X6CrNiTi18-10* **4**—*X2CrNiMoN22-5-3*	**6**—600 °C **8**—800 °C	**0**—slow cooling in the furnace **1**—fast cooling in water mist	**(+20)**—+20 °C **(−20)**—–20 °C

**Table 2 materials-16-03281-t002:** Chemical composition of the tested samples made of the *S355J2+N* steel (according to [8]).

**Element**	**C**	**Si**	**Mn**	**P**	**S**	**Cr**	**Ni**	**Mo**
Content by wt. %	0.185	0.164	1.44	0.0066	0.003	0.0298	<0.005	<0.005
**Element**	**Ti**	**Cu**	**Al**	**Co**	**Nb**	**V**	**W**	
Content by wt. %	<0.002	0.0861	0.0349	0.0032	<0.002	<0.002	<0.015	

**Table 3 materials-16-03281-t003:** Values of the *SFA*_(*n*)_ (%) parameters for the *S355J2+N* steel, determined on impact toughness test fractures obtained during this research.

Sample Number	Temperature of Test (°C)	*SFA*_(1)_ (%)		*SFA*_(2)_ (%)		*SFA*_(3)_ (%)		*SFA*_(4)_ (%)		*SFA*_(5)_ (%)
		Average	Standard Deviation	Average	Standard Deviation	Average	Standard Deviation	Average	Standard Deviation	
1	+20	100		100		100		100		100
	−20	47.07	8.21	54.90	7.22	51.30	7.68	69.25	6.11	50
160	+20	100		100		100		100		100
	−20	55.95	5.90	63.20	4.68	59.91	5.15	75.55	3.77	65
161	+20	100		100		100		100		100
	−20	46.45	4.65	54.20	3.54	50.64	3.97	68.99	3.27	52
180	+20	100		100		100		100		100
	−20	43.85	5.90	54.61	5.87	49.81	5.93	67.23	4.38	52
181	+20	91.62	18.74	93.17	15.28	92.47	16.83	95.27	10.57	100
	−20	44.50	7.97	57.27	6.22	51.72	6.98	68.15	5.74	60

**Table 4 materials-16-03281-t004:** Chemical composition of the samples made of *X20Cr13* steel (according to [8]).

**Element**	**C**	**Si**	**Mn**	**P**	**S**	**Cr**	**Ni**	**Mo**
Content by wt. %	0.247	0.428	0.784	0.0153	0.0166	13	0.1	0.146
**Element**	**Ti**	**Cu**	**Al**	**Co**	**Nb**	**V**	**W**	
Content by wt. %	0.0045	0.0479	0.0086	0.0132	<0.002	0.009	<0.02	

**Table 5 materials-16-03281-t005:** Values of the *SFA*_(*n*)_ (%) parameters determined for *X20Cr13* steel, determined on impact toughness test fractures obtained during this research.

Sample Number	Temperature of Test (°C)	*SFA*_(1)_ (%)		*SFA*_(2)_ (%)		*SFA*_(3)_ (%)		*SFA*_(4)_ (%)		*SFA*_(5)_ (%)
		Average	Standard Deviation	Average	Standard Deviation	Average	Standard Deviation	Average	Standard Deviation	
2	+20	20.19	5.70	36.71	5.55	29.44	5.51	46.98	6.12	11
	−20	7.95	1.28	7.95	1.28	7.95	1.28	28.09	2.39	0
260	+20	14.54	3.20	14.54	3.20	14.54	3.20	37.87	4.39	11
	−20	8.92	1.37	8.92	1.37	8.92	1.37	29.77	2.31	0
261	+20	19.69	4.79	19.69	4.79	19.69	4.79	43.99	5.81	6
	−20	10.54	2.86	10.54	2.86	10.54	2.86	32.19	4.20	0
280	+20	38.99	9.06	38.99	9.06	38.99	9.06	62.04	7.08	6
	−20	16.33	3.68	16.33	3.68	16.33	3.68	40.16	4.53	0
281	+20	16.19	3.92	16.19	3.92	16.19	3.92	39.97	4.54	6
	−20	9.62	2.58	9.62	2.58	9.62	2.58	30.72	4.27	0

**Table 6 materials-16-03281-t006:** Chemical composition of the samples made of *X6CrNiTi18-10* steel (according to [8]).

**Element**	**C**	**Si**	**Mn**	**P**	**S**	**Cr**	**Ni**	**Mo**
Content by wt. %	0.0709	0.467	1.84	0.0246	<0.005	18	9.12	0.347
**Element**	**Ti**	**Cu**	**Al**	**Co**	**Nb**	**V**	**W**	
Content by wt. %	0.352	0.261	0.0356	0.00983	0.0167	<0.087	<0.02	

**Table 7 materials-16-03281-t007:** Values of the *SFA*_(*n*)_ (%) parameters determined for *X6CrNiTi18-10* steel, determined on impact toughness test fractures obtained during this research.

Sample Number	Temperature of Test (°C)	*SFA*_(1)_ (%)		*SFA*_(2)_ (%)		*SFA*_(3)_ (%)		*SFA*_(4)_ (%)		*SFA*_(5)_ (%)
		Average	Standard Deviation	Average	Standard Deviation	Average	Standard Deviation	Average	Standard Deviation	
3	+20	100		100		100		100		100
	−20	100		100		100		100		100
360	+20	100		100		100		100		100
	−20	100		100		100		100		100
361	+20	100		100		100		100		100
	−20	100		100		100		100		100
380	+20	100		100		100		100		100
	−20	100		100		100		100		100
381	+20	100		100		100		100		100
	−20	100		100		100		100		100

**Table 8 materials-16-03281-t008:** Chemical composition of the samples made of *X2CrNiMoN22-5-3* steel (according to [8]).

**Element**	**C**	**Si**	**Mn**	**P**	**S**	**Cr**	**Ni**	**Mo**
Content by wt. %	0.0507	0.266	1.8	0.027	<0.005	23.7	4.74	2.92
**Element**	**Ti**	**Cu**	**Al**	**Co**	**Nb**	**V**	**W**	
Content by wt. %	0.0082	0.184	0.0097	0.0622	0.0056	0.0385	<0.02	

**Table 9 materials-16-03281-t009:** Values of the *SFA*_(*n*)_ (%) parameters determined for *X2CrNiMoN22-5-3* steel, determined on impact toughness test fractures obtained during this research.

Sample Number	Temperature of Test (°C)	*SFA*_(1)_ (%)		*SFA*_(2)_ (%)		*SFA*_(3)_ (%)		*SFA*_(4)_ (%)		*SFA*_(5)_ (%)
		Average	Standard Deviation	Average	Standard Deviation	Average	Standard Deviation	Average	Standard Deviation	
4	+20	100		100		100		100		100
	−20	100		100		100		100		100
460	+20	100		100		100		100		100
	−20	100		100		100		100		100
461	+20	100		100		100		100		100
	−20	100		100		100		100		100
480	+20	100		100		100		100		100
	−20	100		100		100		100		100
481	+20	100		100		100		100		100
	−20	100		100		100		100		100

## Data Availability

The data presented in this article are available within the article.

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
