# Peer review of "Impact Fracture Surfaces as the Indicators of Structural Steel Post-Fire Susceptibility to Brittle Cracking"

_materials, 2023, doi:10.3390/ma16083281_

Round 1
Reviewer 1 Report
This article reviews the ‘Impact fracture surfaces as the indicators of structural steel post-fire susceptibility to brittle cracking’. The importance of the topic is good. However, the following points need to be considered.
- In the introduction section, it is necessary to refer to the works of other researchers and similar research.
- According to what standard was the toughness test done?
- Provide a schematic of the impact test.
- In the last section, present the results case by case.
- SFA should be fully introduced for the first time.
- 4 types of steel are used in this research, each of which represents a type of structure. But these structures are not confirmed by the microstructure images in the manuscript. Therefore, authors need to provide microstructure images.
- Considering that the test was carried out at temperatures above and below zero, why has not been discussed about the transition temperature of softness to the brittleness of steels?
Overall, the most important weakness of this manuscript is the lack of relation between failure and microstructure, which is a very important issue.
Author Response
Dear Reviewer,
Detailed answers of the Authors are included in the attached file.
Yours sincerely,
Mariusz Maslak

Reviewer 2 Report
The paper analyzes the effect of temperature history on the cracking properties (energy, surface facture partition, brittleness) of different steels. The paper should be rejected for suspected plagiarism. Indeed, the novelty with respect to reference [6] is not clear. Results reported in the submitted paper seem identical to the ones reported in [6], and the additional elaboration of information is modest. The paper may also be criticized as a salami publication.
1. The language needs to be deeply and carefully revised. Several words are not appropriate and sentences convoluted. Authors are recommended to ask a native/familiar English speaker to review the document before the submission.
2. Several parts of the document can be shortened. Long descriptions can be effectively substituted by more concise tables or figures.
3. The labeling of samples is concise, but cryptic. A more extended definition of sample properties may be helpful, at least in labels of subfigures representing the fracture surface.
4. Figure 5 looks identical to Figure 2 in reference [6] (only the font changes). Usage of the same figure is allowed, but suitable reference to the source must be reported.
5. Subfigure 7 (b) is identical to Figure 3(a) in reference [6].
6. Figure 1 is equal to Figure 4 in reference [6].
7. Figures 3(a), (b), and (c) are identical to Figures 5, 7 and 6 in [6], respectively.
Author Response

(The authors gave the same response as above.)

Reviewer 3 Report
Journal: Materials
Manuscript ID: materials-2338101
Title: Type of manuscript: Article
Title: Impact fracture surfaces as the indicators of structural steel
post–fire susceptibility to brittle cracking. .
Mariusz Maslak, Michal Pazdanowski, Marek Stankiewicz, Anna
Wassilkowska, Paulina Zajdel, Michał Zielina
Rate the Manuscript:
Significance to field and specialization of “Materials” journal: good.
In the has presented the results of experimental date on forecasting post-fire
resistance to brittle failure of selected steel grades used in construction
are presented and discussed in this paper. It has been shown, that the relationships formulated based on these tests agree well with conclusions drawn based on precise analysis of appropriateF–s curves. Furthermore, other relationships, between lateral expansion LE
and energy Wt required to break the sample, constitute an additional
verification in both qualitative and quantitative terms. These relationships
are accompanied here by values of the SFA(n) parameter, which are different,
depending on the character of the fracture. Steel grades differing in
microstructure have been selected for the detailed analysis, including:
S355J2+N – representative for materials of austenitic – pearlitic
structure, and also stainless steels such as X20Cr13 – of martensitic
structure, X6CrNiTi18-10 – of austenitic structure and X2CrNiMoN22-5-3
duplex steel - of austenitic – ferritic structure.
The main conclusions:
The results presented in this paper confirm the need for detailed analysis of impact strength test fractures obtained experimentally in order to reliably draw conclusions on the possible future safe application of material which has survived a fire incident.
The considered fractures are analyzed a posteriori, i.e. after finished simulated firedents, on the samples made of given steel grade cooled down successfully after prior exposure to high temperature.
It is postulated to include this test in the engineering practice of traditionally performed visual inspection of the deformed structure performed on site accompanied by experimental verification of the basic mechanical properties of affected material. Obtained results should warrant that during prolonged service after a fire no brittle fractures would occur in bearing steel structural components, as these fractures during unstable development may result in structural failure. Sufficiently high impact strength of the material warrants that the possible brittle fractures will be effectively arrested. Reliability of the conclusions drawn based on the approach described in this paper is enhanced by the fact, that the results obtained are confirmed by several methods applied.
Question: Haw about the implementation of the current results to obtained the optimal resolution?
Questions: What is the main question addressed by the research?
The references are appropriate.This research based on 55 scientific works.
Scientific content: good.
Originality: good.
Clarity and presentation: acceptable.
Appropriateness for Journal: appropriate subject matter for the “Materials”
Need for rapid publication: no.
Conclusions consistent with the evidence and arguments
presented and they address the main question posed and practically similar to abstract. Please consider the next papers in which take in to account the impact fracture surfaces and value as the indicators of structural steel degradation
: Reliability of steam pipelines of thermal power plants in the course of long-term operation // Materials Science (Springer).– 2006, No 4, vol.42.- P. 421 –424. https://doi.org/10.1007/s11003-006-0101-x; Workability Assessment of Structural Steels of Power Plant Units in Hydrogen Environments: Strength of Materials - 2009, vol. 41, - No 1. - P. 52-57. DOI: 10.1007/s11223-009-9097-4; Strength of welded joints of Cr-Mn steels with elevated content of nitrogen in hydrogen-containing media // Materials Science (Springer).– 2009, N 1, p. 97-107. https://doi.org/10.1007/s11003-009-9166-7

Author Response

(The authors gave the same response as above.)

Round 2
Reviewer 1 Report
The authors' explanations were convincing and they revised the manuscript well. Its publication is recommended.
Reviewer 2 Report
The reviewer is still concerned about salami pubblication